# An Inulin-Type Fructan CP-A from *Codonopsis pilosula* Alleviated 5-Fluorouracil-Induced Intestinal Mucositis via the ERK/MLCK/MLC2 Pathway and Regulation of Gut Microbiota

**DOI:** 10.3390/ph17030297

**Published:** 2024-02-26

**Authors:** Jiangtao Zhou, Deyun Li, Jiajing Wang, Zhuoyang Cheng, Changjian Wang, Xuepeng Zhang, Xiexin Xu, Jianping Gao

**Affiliations:** 1School of Pharmacy, Shanxi Medical University, Taiyuan 030001, China; zjt881206@sxmu.edu.cn (J.Z.); ldyun523@outlook.com (D.L.); wjj00527@163.com (J.W.); chengzy0328@163.com (Z.C.); wcj970919@outlook.com (C.W.); 15690264960@163.com (X.Z.); xuxiexin0401@163.com (X.X.); 2Medicinal Basic Research Innovation Center of Chronic Kidney Disease, Ministry of Education, Shanxi Medical University, Taiyuan 030001, China; 3Shanxi Provincial Key Laboratory of Drug Synthesis and Novel Pharmaceutical Preparation Technology, Shanxi Medical University, Taiyuan 030001, China

**Keywords:** inulin-type fructan CP-A, intestinal mucositis, inflammation, ERK/MLCK/MLC2 signaling pathway, mucosal barrier, intestinal microbiota

## Abstract

Intestinal mucositis (IM) is a common adverse effect of chemotherapy, limiting its clinical application. *Codonopsis pilosula*-derived CP-A (an inulin-type fructan) is an edible Chinese medicine with anti-inflammatory and gastrointestinal protective effects, which may be useful for treating IM. Here, we explored CP-A’s role in ameliorating IM induced by 5-fluorouracil (5-FU) and investigated the underlying mechanism using in vitro experiments and rat models. Western blotting, immunohistochemistry (IHC), and real-time PCR (RT-PCR) analyses were used to assess protein expression related to the extracellular-regulated protein kinases (ERK)/myosin light chain kinase (MLCK)/myosin light chain 2 (MLC2) signaling pathway and tight junction proteins. Inflammatory factors were quantified using enzyme-linked immunosorbent assays (ELISAs), and 16S rRNA amplicon sequencing was employed for cecum content analysis. The results indicated that CP-A restored body weight and food intake and reversed histopathological changes in IM rats. Further, abnormal MLCK activation induced by 5-FU was attenuated by CP-A via the ERK/MLCK/MLC2 pathway. CP-A treatment improved tight junction protein levels and reduced inflammatory factor expression. Moreover, CP-A intervention regulated the intestinal microbiota community structure, increasing the abundance of *Lactobacillus* and decreasing the abundance of *Shigella*. In conclusion, CP-A mitigates 5-FU-induced IM by inhibiting the ERK/MLCK/MLC2 pathway, reducing the expression of inflammatory factors, improving the intestinal mucosal barrier, and regulating the intestinal microbial community. This study highlights CP-A’s therapeutic potential in IM treatment and provides insights for future research.

## 1. Introduction

Many cancers, including stomach, esophageal, breast and head cancer, are addressed with 5-FU treatment [1,2], and it is the first-line treatment for colorectal cancer [3,4]. However, the use of 5-FU can result in severe adverse reactions, such as IM [5]. The clinical manifestations of IM include diarrhea, nausea and vomiting, which cause intense discomfort to patients and seriously limit the use of chemotherapeutic drugs [6,7]. Unfortunately, no effective medications for alleviating the pain associated with IM currently exist; therefore, identifying and developing treatments that can effectively alleviate the symptoms is necessary for enhancing the quality of life for patients under-going chemotherapy. 5-FU exhibits cytotoxic effects on intestinal epithelial cells, resulting in damage to the intestinal mucosal barrier and the release of pro-inflammatory factors [8]. Tight junction proteins are essential components of the intestinal barrier [9]. 5-FU reaches the intestinal mucosal barrier primarily by degrading tight junction proteins [10]. In addition, 5-FU disrupts the intestinal microbiota [11,12]. These factors offer potential avenues for addressing IM.

Pro-inflammatory factors play an important role in the occurrence and development of inflammation [13,14]. Overexpression of tumor necrosis factor-alpha (TNF-α) and interferon gamma (IFN-γ) is often detected in cases of IM [15,16]. ERK and MLCK play central roles in the protection of intestinal epithelial tight junction [17]. By promoting the phosphorylation of MLC2, MLCK is activated by Phospho-ERK1/2, leading to the destruction of intestinal tight junction and degradation of tight junction proteins such as Claudin-1, Occludin, F-actin and Zonula occludens-1 (ZO-1) [18,19]. TNF-α activation of ERK1/2 promotes ERK phosphorylation, thus activating the ERK/MLCK/MLC2 pathway [19]. In addition, the overexpression of IFN-γ is involved in the activation of MLCK, which up-regulates the expression of Claudin-2 and promotes interleukin (IL)-13 release [20]. Once the intestinal barrier is breached, an increase in intestinal epithelium permeability permits endotoxins and macromolecules to enter the cells freely, causing the release of IL-6, IL-8, TNF-α and IFN-γ, which will aggravate the inflammatory response [18]. Moreover, the destruction of the intestinal mucosal barrier is accompanied by disturbances in the intestinal microbiota [9,21]. Therefore, the inhibition of TNF-α and IFN-γ overexpression and the regulation of the ERK/MLCK/MLC2 pathway protect the intestinal mucosal barrier and reduce inflammation. A stable and healthy microbiome is essential for the proper functioning of the gut [8]. Many studies have indicated that chemotherapy-induced gastrointestinal diseases are often accompanied by disturbance of intestinal microbiota [10]. Dysregulation of intestinal microbiota leads to disruption of intestinal homeostasis and intestinal-mucus-barrier integrity [9,11]. There is increasing evidence that 5-FU-induced IM can cause the dysbiosis of intestinal microbiota, in which the abundance of harmful bacteria increases, and the abundance of probiotics decreases [10,12]. Thus, regulating intestinal microbiota and increasing the abundance of probiotics may be a potential treatment for 5-FU-induced IM.

Traditional Chinese medicine (TCM) and natural products extracted from TCM have a long history of clinical application in the treatment of various diseases, including IM [22]. *Codonopsis pilosula* is an edible herbal medicine widely used for improving poor gastrointestinal function in TCM. The medicine is derived from the roots of *Codonopsis pilosula* (Franch.) Nanf., *Codonopsis pilosula* (Franch.) Nannf. var. *modesta* (Nannf.) L.T. Shen and *Codonopsis tangshen* Oliv. Recently, the edible plant has been listed by the National Health Commission of China as a substance that is both food and traditional Chinese medicine. Over the past several years, *Codonopsis pilosula*, a plant homologous to food and medicine, has become known for its anti-ulcerative colitis effects, its role in regulating intestinal microbiota, and its ability to decrease inflammatory factor levels [23,24]. The main active constituents of *Codonopsis pilosula* are polysaccharides, lignans and polyacetylenes [25]. Additionally, the anti-inflammatory effects of polysaccharides from *Codonopsis pilosula* have been reported [24,26]. CP-A represents the most active ingredient in *Codonopsis pilosulae* polysaccharides. Nevertheless, research on IM treatment is rare. Meanwhile, its in vivo and in vitro anti-IM mechanisms remain unclear. Hence, this study aimed to investigate the mechanisms underlying CP-A treatment for 5-FU-induced chemotherapeutic enteritis using cell cultures and rat experiments. This study will provide valuable insights for the development of improved treatments for IM, addressing a critical need in clinical oncology.

## 2. Results

### 2.1. Cytotoxic Effects of CP-A on IEC-6 Cells

IEC-6 cells were treated with varying concentrations of CP-A for either 24 or 48 h, and no significant cytotoxicity was observed based on the MTT assay results, as shown in Figure 1b. Low and high doses of CP-A at 100 μg/mL and 200 μg/mL, respectively, were selected for subsequent in vitro studies.

### 2.2. Protective Effects of CP-A on 5-FU-Induced Intestinal Mucositis in Rats

The study established a rat model that involved a continuous intraperitoneal injection of 5-FU for 3 days. The experimental design is displayed in Figure 2a. Distinct from the control group, rats in the other groups experienced weight loss from the second day. Following BTC or CP-A administration, this trend was reversed. Rats in the BTC and CP-A medium- and high-dose groups gradually gained weight from the fifth day, while those in the CP-A low-dose group gained weight from the sixth day. In contrast, rats in the control group continued to gain weight throughout the experiment, while those in the 5-FU group continued to lose weight starting on the second day (Figure 2b). Food intake and diarrhea showed similar patterns of changes (Figure 2c,d). Histopathological results revealed varying degrees of damaged intestinal epithelium caused by 5-FU. The rats in the 5-FU group exhibited the most severe intestinal damage, with significant mucosal necrosis observed, compared to those in the control group. Administration of BTC and CP-A resulted in a dose-dependent alleviation of intestinal damage, as depicted in Figure 2e, while the histopathological scores showed a notable reduction compared to the group administered with only 5-FU (*p* < 0.01), as presented in Figure 2f.

### 2.3. Effect of CP-A on Inflammatory Cytokines

Assays conducted on IEC-6 cells revealed that 5-FU administration upregulated TNF-α, IFN-γ, IL-6, IL-8 and IL-13 expression levels. Conversely, IL-4 and IL-10 levels decreased significantly. However, BTC and CP-A interventions reversed this trend, especially at CP-A concentrations of 200 μg/mL and 60 mg/kg (*p* < 0.01) (Figure 3).

### 2.4. Effect of CP-A on ERK/MLCK/MLC2 Signaling Pathway

After demonstrating that CP-A improved inflammatory cytokines levels, we conducted a Western blot analysis to assess changes in the ERK/MLCK/MLC2 pathway following CP-A administration. We detected various proteins (ERK1/2, p-ERK1/2, MLCK, MLC2 and p-MLC2) associated with the ERK/MLCK/MLC2 pathway in both IEC-6 cells and small-intestinal tissues. As shown in Figure 4a–d,f–i, p-ERK1/2 was activated in the 5-FU group, and this activation was diminished after CP-A administration (*p* < 0.01). MLCK, which is downstream of p-ERK1/2, was significantly downregulated compared to the group administered only 5-FU (*p* < 0.01). Western blot results obtained from rats treated with 5-FU and IEC-6 cells treated with 5-FU were similar. Protein immunohistochemical staining also yielded results consistent with those of the Western blot analysis (Figure 4e,j).

### 2.5. Effect of CP-A on Intestinal Mucosal Barrier Proteins

Western blotting was performed to determine the expression of tight junction proteins in IEC-6 cells, while IHC staining was conducted to detect the protein expression in the small-intestine tissues. The results proved that 5-FU significantly downregulated the expression of Occludin, ZO-1 and Claudin-1 and upregulated the expression of Claudin-2. Nevertheless, these effects were reversed by CP-A. Similarly, 5-FU remarkably reduced the concentrations of Occludin, ZO-1, Claudin-1 and F-actin, compared to the control group. After CP-A and BTC administration, the concentrations of these tight junction proteins increased (Figure 5a–f). These findings demonstrate the significant role that CP-A plays in treating cellular and intestinal tissue barrier dysfunction.

Then, RT-PCR was used to assess the mRNA levels of MLCK, ERK1/2, TNF-α, IL-4, IL-6 and IL-10. In the 5-FU group, a significant increase in the expression of MLCK, TNF-α and IL-6 was noticed, as compared with the control group. However, the expression of ERK1/2, IL-4 and IL-10 showed the opposite pattern. The RT-PCR results were similar to those of ELISA, Western blotting and immunohistochemical staining (Figure 5g–l).

### 2.6. Effect of CP-A on Microbial Diversity

The Shannon index rarefaction curve reflects the measure of species richness and diversity in the sample (Figure 6a). The Venn diagram shows the number of ASVs common and unique to each group (Figure 6b). Overlap analysis showed that 549 ASVs were present in all six groups. In addition, 8996, 9081, 8513, 7989, 8431 and 6856 unique ASVs were observed in the control, 5-FU, BTC and CP-A (L, M, and H) groups, respectively. To evaluate the richness and diversity of the microbiota community, we utilized the Chao 1, Shannon, and Simpson indices for alpha diversity analysis (Figure 6c). The rats in the 5-FU group had the highest Chao 1, Shannon, and Simpson indices. After CP-A treatment, these indices decreased. These results suggested that 5-FU increased the diversity of the microbial community. Beta diversity was used to measure the similarity of microbial communities in each group. PCoA based on the Bray–Curtis distance showed the largest differences between the control, 5-FU and high-dose CP-A groups (Figure 6d), indicating that CP-A treatment likely influenced community structure.

### 2.7. Taxonomic Composition Analysis of Microbial Community

The gut microbial composition was determined at the Phylum, Class, Order, Family and Genus levels. At the Phylum level, *Firmicutes, Proteobacteria, Bacteroidetes* and *Actinobacteria* were the most abundant, and their abundance values were more than 95% (Figure 7a and Appendix A). After 5-FU injection, the abundance of *Firmicutes* and *Bacteroidetes* decreased from 89.49% to 62.86% and from 6.97% to 6.54%, respectively. In addition, the abundance values for *Proteobacteria* and *Actinobacteria* increased to 27.03% and 2.57%, respectively, after 5-FU treatment. However, CP-A intervention restored their abundance to normal levels. 5-FU treatment increased the Firmicutes/Bacteroidetes (F/B) ratio, which was significantly restored by 60 mg/kg CP-A treatment.

At the class level, *Bacilli, Clostridia, Bacteroidia* and *Erysipelotrichi* were the most abundant in the control group (Figure 7b and Appendix A). The abundance of *Gammaproteobacteria* was the second highest value in the 5-FU group (*p* < 0.01 vs. control and CP-A H). After 5-FU administration, the abundance of *Bacilli* decreased notably (*p* < 0.01), whereas that of *Clostridia* increased to 43.28%, making it the highest value in the 5-FU group. At the Order level, *Lactobacillales, Clostridiales, Bacteroidales* and *Erysipelotrichales* were the most abundant in the control rats, whereas *Clostridiales* and *Enterobacteriales* were the two most abundant in the 5-FU rats (Figure 7c and Appendix A). Furthermore, the abundance of *Enterobacteriales* increased markedly in the 5-FU group (*p* < 0.01 vs. control and CP-A H). *Lactobacillaceae* were the dominant bacteria in the control group and *Enterobacteriaceae* were the dominant bacteria in the 5-FU group, at the Family level (Figure 7d and Appendix A). Post CP-A and BTC treatment, *Enterobacteriaceae* decreased significantly (*p* < 0.05). *Lactobacillus* was characteristic of the control group, whereas *Shigella* dominated the 5-FU group, at the genus level. *Lactobacillus* decreased significantly in the 5-FU group compared to the control group (*p* < 0.01). After CP-A and BTC treatment, *Lactobacillus* levels were markedly increased (*p* < 0.01) (Figure 7e and Appendix A).

### 2.8. LEfSe Analysis of Microbial Community

LEfSe analysis was used to identify biomarkers in each group (Figure 8a,b). In the control group, marker bacterial genera included *g_Prevotella*, *g_Streptococcus*, *g_Acetobacter* and *g_Ralstonia*. Conversely, *g_Olsenella* and *g_Bacteroides* were highly expressed in the 5-FU group. Following BTC and CP-A administration, *g_Caulobacter* was the main bacterium in the BTC group, while *g_Lactobacillus* and *g_Veillonella* were the marker bacteria in the CP-A H group, *g_Corynebacterium*, *g_Macrococcus*, *g_Facklamia* and *g_Oligella* were the marker bacteria in the CP-A L group and *g_Allobaculum*, *g_Coprococcus* and *g_Wohlfahrtiimonas* were the marker bacteria in the CP-A M group.

## 3. Discussion

The main manifestations of IM induced by 5-FU were diarrhea and decreased food intake [27], which are primarily attributed to the damage inflicted upon the integrity of the intestinal mucosa by 5-FU [28]. Previous research has indicated that the mucosa lining of the small intestine is particularly susceptible to the detrimental effects of 5-FU treatment [29,30]. Our study aimed to investigate the effect of CP-A on inflammation induced by 5-FU, both in vitro and in vivo. Our findings demonstrated that CP-A effectively restored body weight and food intake, alleviated diarrhea symptoms and reduced intestinal mucosal damage in rats with 5-FU-induced IM. These results suggest that CP-A has an inhibitory effect on 5-FU-induced IM and probably plays a role in inhibiting the ERK/MLCK/MLC2 signaling pathway and regulating microbial structure.

Inflammation can occur following the release of pro-inflammatory factors and often accompanies various immune responses and processes [31,32]. TNF-α, an inflammatory cytokine, is produced by macrophages and monocytes during inflammation, and it is involved in the conduction of various intracellular signals leading to cell necrosis or apoptosis [33]. Additionally, IFN-γ is another essential pro-inflammatory cytokine that is released during various inflammatory conditions [34,35]. Endotoxins and exotoxins are the most common inducers of cytokine production [36]. As inflammation progresses, the generation of inflammatory cytokines, including TNF-α and IFN-γ, orchestrates the innate immune response against infections [37]. However, excessive expression of TNF-α and IFN-γ is harmful, and IL-6 and IL-8 also contribute to inflammation [38]. Furthermore, both IL-4 and IL-10 were found to downregulate MCP-1 production, which plays a crucial role in intestinal inflammation, not only in Caco2 cells but also in freshly isolated epithelial cells [39]. Our in vitro and in vivo experiments revealed that CP-A exhibited remarkable anti-inflammatory effects by modulating inflammatory cytokine levels in IEC-6 cells stimulated with 5-FU and in rats with IM. CP-A significantly reduced the levels of IL-6, IL-8, TNF-α and IFN-γ, while enhancing IL-4 and IL-10 expression in both cell cultures and the rat models.

Tight junction proteins play a protective role in the intestinal mucosal barrier, and their degradation is often observed during intestinal inflammation [9]. An increasing number of studies have shown that the degradation of tight junction proteins is accompanied by the activation of ERK and MLCK [17,20]. MLCK promotes the phosphorylation of MLC2, which increases the expression of p-MLC2 and produces ATP, which degrades tight junction proteins [34]. Moreover, a 5-FU-induced increase in TNF-α and IFN-γ expression not only promotes the occurrence of inflammation but also acts as an activator of ERK and MLCK [9]. Furthermore, the activation of MLCK leads to increased secretion of IL-13 and to Claudin-2 expression [20,40]. The ERK/MLCK/MLC2 signaling pathway becomes activated due to elevated p-ERK1/2 levels and MLC2 phosphorylation following 5-FU administration in both cellular and rat models. However, treatment with CP-A significantly reduced p-ERK1/2 levels, along with MLC2 phosphorylation and MLCK expression, thereby deactivating the ERK/MLCK/MLC2 signaling pathway. Thus, Claudin-1, F-actin, Occludin and ZO-1 expression levels were upregulated in the IEC-6 cells and rat small intestines, whereas claudin-2 and IL-13 expression levels were downregulated. Increased levels of tight junction protein in the small intestine restored intestinal mucosal barrier function and repaired intestinal damage. Therefore, CP-A restored the expression of tight junction protein by regulating the ERK/MLCK/MLC2 signaling pathway to restore intestinal mucosal barrier function and reduce the intestinal damage caused by 5-FU.

To assess the potential cytotoxicity of CP-A on IEC-6 cells, we performed an MTT assay, which indicated that CP-A promotes cell growth without causing cytotoxic effects. Another study also reported that CP-A does not have cytotoxic effects [41]. Western blot and IHC results revealed that CP-A significantly restores Claudin-1, Occludin and ZO-1 levels, while down-regulating IL-6, IL-8, TNF-α and IFN-γ.

Additionally, the gut microbiota plays a crucial role in maintaining intestinal homeostasis, as it acts as an ecological barrier [42]. Dysbiosis within the gut microbiota can lead to various adverse effects, such as diarrhea and indigestion, and is associated with inflammatory bowel diseases, such as ulcerative colitis or Crohn’s disease [4,43]. Interestingly, our sequencing data revealed that 5-FU disrupted the overall structure and abundance of the gut microbiota and altered the relative abundance of specific bacterial genera known to inhabit the intestine [11,12,21]. The results of α-diversity analysis indicated that high-dose CP-A decreased the richness and diversity of the microbiota. However, the results of β-diversity analysis indicated that the high-dose CP-A group and the control group had similar microbiota structures, whereas the 5-FU group displayed a significantly distinct microbiota structure. Hence, CP-A effectively stabilized the microbial community.

*Firmicutes*, *Proteobacteria*, *Bacteroidetes* and *Actinobacteria* are the four dominant bacterial phyla found in the guts of normal organisms, with *Firmicutes* having the highest relative abundance. The ratio of Firmicutes/Bacteroidetes (F/B) correlates with the extent of intestinal inflammation, and a higher proportion is found in 5-FU-induced intestinal mucositis. Our findings are consistent with those of previous studies showing that 5-FU can alter the gut microbiota composition by decreasing *Firmicutes* abundance while increasing *Proteobacteria* prevalence. This shift in composition may affect dietary energy absorption, given *Firmicutes*’s role in this process [44]. Current research has shown that dysregulation of the gut microbiota accompanies 5-FU-induced IM, leading to an increased abundance of *Proteobacteria* abundance and gastrointestinal disorders such as enteritis and diarrhea 4. Interestingly, CP-A reversed these changes in the abundance of Firmicutes and Proteobacteria, offering a potential avenue for mitigation.

To further investigate bacterial genera specifically altered by 5-FU and CP-A treatment, we examined their relative abundance levels among the rats used in this study. 5-FU significantly increased the relative abundance of *Shigella* and decreased the relative abundance of *Lactobacillus*; however, CP-A restored the relative abundance of these genera. Notably, high-dose CP-A treatment resulted in a significant increase in the relative abundance of *Lactobacillus* compared to levels in the other groups. *Lactobacillus* is an intestinal probiotic that effectively combats acute diarrhea and intestinal mucosal lesions, and is the dominant bacteria in the normal intestine, while *Shigella* is almost absent [45,46,47]. However, after the occurrence of lesions, the relative abundance of *Shigella* increases, and the advantages of *Lactobacillus* diminish substantially [47,48]. *Shigella* is a small gram-negative bacterium that invades the intestinal epithelium and causes shigellosis, also known as bacillary dysentery [48,49]. The main cause of bacterial diarrhea is *Shigella* infection, which is highly lethal in malnourished people [50,51]. Therefore, CP-A administration was found to alleviate 5-FU-induced intestinal mucositis in rats by reducing the prevalence of *Shigella* and increasing the prevalence of *Lactobacillus*. Thus, *Lactobacillus* may be a beneficial genus in the treatment of IM, although the probiotic mechanism of CP-A requires further research.

## 4. Materials and Methods

### 4.1. Chemicals and Reagents

CP-A (Figure 1a, MW: 3.6 kDa), an inulin-type fructan, was isolated from *Codonopsis pilosula*, as previously described [52]. 5-FU (F6173) was purchased from Macklin Biochemical Co., Ltd. (Shanghai, China). Bifid. Triple Viable Capsules Dissolving at Intestines (BTC) were purchased from Jincheng HEALTH Pharmaceutical Co., Ltd. (Jincheng, China). Primary antibodies against rat MLC2 (DF7911 1:2000) and Phospho-MLC2 (Ser15 AF8618 1:1000) were purchased from Affinity Biosciences Co., Ltd. (Cincinnati, OH, USA). MLCK (21642-1-AP 1:1000), ERK1/2 (11257-1-AP 1:5000), Phospho-ERK1/2 (Thr202/Tyr204 28733-1-AP 1:2000), Beta Actin (20536-1-AP, 1:2000), Claudin-2 (26912-1-AP, 1:1000), Occludin (27260-1-AP, 1:5000) and ZO-1 (21773-1-AP, 1:5000) were purchased from Proteintech Group, Inc. (Wuhan, China). Claudin-1 (bs-1428R, 1:1000) and F-actin (bs-1571R) were obtained from Beijing Biosynthesis Biotechnology Co., Ltd. (Beijing, China). The secondary antibodies were obtained from ImmunoWay Biotechnology (Plano, TX, USA). All ELISA kits were acquired from Jiangsu Meimian Industrial Co., Ltd. (Yancheng, China). All of the other materials were of analytical grade.

### 4.2. Cells Culture and MTT Assay

The rat intestinal epithelial cell line IEC-6 was acquired from the BeNa Culture Collection (BNCC100548) and maintained in complete high-glucose DMEM culture medium (Thermo Fisher Gibco, Waltham, MA, USA, Cat. No. C11995500BT) with 10% fetal bovine serum (FBS, AusGeneX, Molendinar, Australia, Cat. No. 10100147) and 1% streptomycin/penicillin (Gibco, New York, NY, USA, Cat. No. 15240-062) at 37 °C and in 95% air and 5% CO_2_. Cell viability was assessed using the 3-(4,5-dimethylthiazol-2-yl)-2,5-diphenyltetrazolium bromide (MTT) assay after treatment with different concentrations of CP-A (0, 1, 5, 10, 20, 50, 100, 200 and 500 μg/mL) for 24 h. Briefly, IEC-6 cells (2.0 × 10^3^ cells/well) in the logarithmic (log) growth phase were inoculated into 96-well plates (100 μL/well) and made to adhere for 24 h. Thereafter, the medium was substituted for with CP-A and treated for 24 h. Then, the different concentrations of CP-A were substituted for with MTT (100 μL) in each well at a final concentration of 5 mg/mL and incubated for 4 h. Subsequently, the medium was removed and Dimethyl sulfoxide (DMSO, 100 μL) was added to each well. After shaking the plates for 10 min, cell viability was evaluated at 570 nm. The MTT assay was repeated three times to ensure data reproducibility.

### 4.3. Total RNA Extraction and Quantitative Real-Time PCR

IEC-6 cells were seeded in 6-well plates (2.0 × 10^5^ cells/well) and cultured over 24 h to ensure they adhered to the plate. Cells were treated with different concentrations of CP-A for 1 h and then incubated with or without 5-FU (10 μg/mL) for 24 h [53]. Total RNA was extracted from IEC-6 cells using *TransZol Up* reagent (Transgen, Beijing, China, Cat. No. ET111-01). RNA concentration was detected using an Eppendorf Bioluminometer D30 (Roche, Mannheim, Germany). Subsequently, a cDNA Synthesis Supermix kit (Transgen, Beijing, China, Cat. No. AT311-02) was used to reverse transcribe total RNA into cDNA according to the manufacturer’s instructions. Ultimately, each RT-PCR reaction system volume was 20 µL, which consisted of 10 µL 2 × Tip Green qPCR SuperMix (Transgen, Cat. No. AQ141-01), 0.4 µL Forward and Reverse Primers (10µM) for each gene [Appendix A], 6.2 µL of cDNA template (diluted 10-fold using Nuclease-free Water) and 3 µL Nuclease-free Water. The RT-PCR analyses were performed in a Light Cycler^96^ Real-Time PCR System (Roche, Mannheim, Germany). The experiment was conducted according to the indicated reaction program (one cycle of 94 °C for 30 s, followed by 40 cycles of 94 °C for 5 s and 60 °C for 30 s). The primer sequences for the target genes, such as MLCK, ERK1/2, TNF-α, IL-4, IL-6 and IL-10, were designed using Primer-BLAST software (www.ncbi.nlm.nih.gov/tools/primer-blast, accessed on 5 October 2023) as shown in Appendix A. The relative quantification of all mRNA expression was performed using the 2^−ΔΔCt^ method. GAPDH was purchased from Sangon Biotech (Shanghai, China, Cat. No. B661204-0001) as an internal reference for the normalization of the target genes.

### 4.4. Experimental Animals

Male SD rats [SPF grade, weighing 200 ± 20 g, approval number: SCXK (Jing) 2019-0010] were provided by Beijing Sipeifu Biotechnology Co., Ltd., Beijing, China. All animals used in this study were raised on a 12 h light/dark cycle and allowed free access to food and clean water under regulated conditions, where room temperature was maintained between 20 °C and 24 °C with relatively constant humidity. All experimental protocols were approved by the Animal Experimental Ethics Committee of the Shanxi Medical University. In accordance with the regulations on human care and use of laboratory animals, all reasonable measures were taken to minimize animal suffering.

### 4.5. Establishment of 5-FU-Induced Intestinal Mucositis

All rats were allowed to acclimate for a week and then randomly divided into six groups (n = 12): control group, 5-FU group (50 mg/kg, intraperitoneal injection, i.p.), BTC group (250 mg/kg, intragastric administration, i.g.) and CP-A groups (15, 30, and 60 mg/kg, i.g.). Over the first 3 days, in the control group, rats were intraperitoneally injected with 0.8 mL of 0.9% saline solution per 200 g of body weight, while the other group rats were administered 5-FU. The vehicle, BTC, or differing doses of CP-A were orally administered to all rats daily during the experiment (Figure 2a). Rats were deeply anesthetized with 10% chloral hydrate 24 h after the final treatment. Blood samples were obtained from the abdominal aorta for biochemical analysis, and the cecal contents were collected to analyze the intestinal microbiota. Small-intestinal tissues were rapidly harvested and rinsed with cold PBS. Ileum samples of all rats (0.5 × 0.5 cm) were fixed in 4% paraformaldehyde and embedded in paraffin for subsequent experiments, while other intestinal tissues were stored at −80 °C for biochemical analysis.

### 4.6. Histopathological Evaluation

Paraffin blocks containing tissue sections of 5-μm-thickness were cut and stained with hematoxylin and eosin (HE) to evaluate ileum tissue damage using an Olympus light microscope (Tokyo, Japan).

### 4.7. Immunohistochemical Staining

IEC-6 cells were inoculated into clean microscope cover-glass slips placed inside 6-well plates (2.0 × 10^5^ cells/well), followed by treatments as mentioned above. The glass plates containing the cells were removed for immunohistochemical analysis.

Paraffin sections and slides of the cells were dewaxed. After blocking endogenous peroxidase activity, sections were incubated overnight at 4 °C with ZO-1, Occludin, Claudin-1, F-actin, MLCK and Phospho-MLC2 antibodies, followed by the addition of secondary anti-rabbit antibodies. This was followed by 1 h incubation indoors. DAB staining was performed after 40 min, followed by restraint using hematoxylin. Images were captured using a Nikon Eclipse Ci scanning microscope (Nikon Instruments, Melville, NY, USA).

### 4.8. ELISA Analysis

Fresh blood samples were centrifuged to collect serum (1000 rpm for 10 min at 4 °C), which the IEC-6 cells were seeded onto six-well plates, as previously described. The cell supernatant was collected for analysis. Serum and IEC-6 supernatant levels of TNF-α, IL-4, IL-6, IL-8, IL-10, IL-13 and IFN-γ were measured using ELISA according to the manufacturer’s instructions.

### 4.9. Western Blotting Experiments

Total protein from the cell samples or small-intestine tissues was extracted using RIPA Lysis buffer (containing 10 μL 100× PMSF, 10 μL phosphatase inhibitor per 1mL) via ultrasonic crusher (Scientz, Ningbo, China) and a tissue grinder (Servicebio, Wuhan, China). The BCA Protein Assay Kit (Keygen Biotech, Nanjing, China, Cat. No. KGP902) was used to quantify the protein concentration. Equal amounts of protein were separated using electrophoresis on sodium dodecyl sulfate-polyacrylamide gels (SDS-PAGE) and transferred onto polyvinylidene difluoride (PVDF) membranes. The membranes were then blocked with a rapid blocking buffer for 15 min before being incubated overnight at 4 °C with various antibodies, including MLC2, Phospho-MLC2, MLCK, ERK1/2, Phospho-ERK1/2, ZO-1, Occludin, Claudin-1, Claudin-2 and β-actin. Finally, the membranes were incubated with secondary antibodies for 1 h. The membranes were washed three times with TBST solution for 5 min each time, followed by incubation with primary and secondary antibodies. The Bio-Rad ChemiDoc XRS imaging system was used to detect protein band intensity after the addition of Electro-Chemi-Luminescence (ECL) reagent (Seven Beijing, China). The analysis was conducted using Image Lab Software version 6.0 (Bio-Rad Laboratories, Hercules, CA, USA).

### 4.10. Gut Microbiota Analysis

Sequencing was performed by Shanghai Personal Biotechnology Co., Ltd. (Shanghai, China). Briefly, total genomic DNA was isolated from the cecal contents using the OMEGA Soil DNA Kit (M5635-02) (Omega Bio-Tek, Norcross, GA, USA), according to the manufacturer’s instructions. The V3–V4 regions of the 16S rRNA were amplified using forward primer 338F (5′-ACTCCTACGGGAGGCAGCA-3′) and reverse primer 806R (5′-GGACTACHVGGGTWTCTAAT-3′). Microbiome bioinformatics analysis was performed with QIIME2 [54]. Briefly, the raw sequence data were demultiplexed using the demux plugin and then primer cut using the cutadapt plugin. The DADA2 plugin was then used for quality filtering, denoising, merging, and chimera removal of the sequences. QIIME2 and R packages (v3.2.0) were utilized for processing sequence data, while ASV-level alpha diversity indices such as Chao1, Shannon, and Simpson were calculated via ASV table analysis. Beta diversity analysis [Bray–Curtis metrics via principal coordinate analysis (PCoA)] showed the structural variation of microbial communities in each sample group, while LDA Effect Size (LEfSe) detected differentially abundant taxa across groups using default parameters.

### 4.11. Data Analysis

All study outcomes were presented as the mean ± standard error of mean (SEM) from three independent experiments. Statistical analyses were performed using SPSS software (version 26.0; IBM, New York, NY, USA) and GraphPad Prism software (version 8.0; San Diego, CA, USA). Student’s *t*-test was used to compare two groups, while one-way analysis of variance (ANOVA) was employed for multiple comparisons of three or more groups, with *p*-values (*p* < 0.05) considered statistically significant.

## 5. Conclusions

These results suggested that CP-A may exert its beneficial effects by reducing inflammatory factors, improving intestinal mucosal barrier integrity, and modulating the gut microbiota. Specifically, CP-A inhibits MLCK expression and suppresses TNF-α and IFN-γ signaling pathways, thus preventing the degradation of Claudin-1, F-actin, Occludin and ZO-1 proteins while also reducing Claudin-2 and IL-13 expression levels. CP-A reduces pro-inflammatory cytokine levels (IL-6, IL-8, TNF-α and IFN-γ), increases anti-inflammatory cytokine levels (IL-4 and IL-10), stabilizes microbial structure, and balances microbial communities by up-regulating *Lactobacillus* relative abundance while down-regulating *Shigella* relative abundance. In conclusion, our work provided a potential novel strategy for treatment of chemotherapy-induced IM using *Codonopsis pilosula*. Moreover, our study provides experimental evidence supporting the potential application of CP-A in patients suffering from chemotherapy-induced mucositis. However, it is important to acknowledge the limitations of this study, which focused only on investigating the molecular mechanism and gut microbiota changes in rats. Future research should comprehensively investigate the anti-IM effect of CP-A, including experimentation on genetically modified animals and fecal bacteria transplantation. These future studies should prioritize the elucidation of the mechanism of action and the establishment of CP-A as a promising candidate drug for the treatment of chemotherapy-induced mucositis.

## Figures and Tables

**Figure 1 pharmaceuticals-17-00297-f001:**
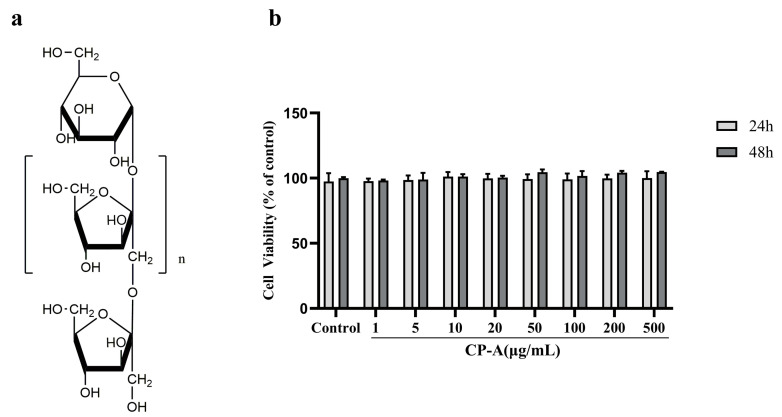
Cytotoxicity of CP-A to IEC-6 cells: (**a**) The chemical structure of CP-A. (**b**) Cell viability was assessed using the 3-(4,5-dimethylthiazol-2-yl)-2,5-diphenyltetrazolium bromide (MTT) assay. Values are represented as the mean ± SEM (n = 3).

**Figure 2 pharmaceuticals-17-00297-f002:**
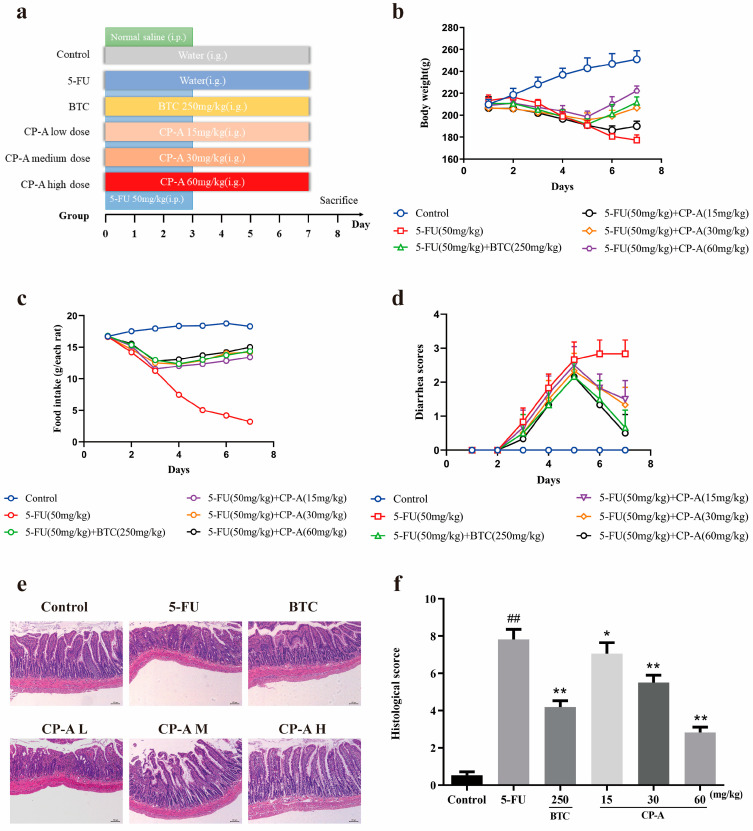
CP-A alleviates the macroscopic manifestation of intestinal mucositis in rats. (**a**) Experimental design and study grouping; (**b**) Body weight changes (n = 6); (**c**) Food intake (n = 6); (**d**) Diarrhea scores (n = 6); (**e**) HE-stained sections from ilea (n = 3, magnification 50×); (**f**) Histopathological scores. CP-A L, low dose of CP-A (15 mg/kg, i.g.); CP-A M, medium dose of CP-A (30 mg/kg, i.g.); CP-A H, high dose of CP-A (60 mg/kg, i.g). Values are represented as mean ± SEM. ## *p* < 0.01 versus control group, * *p* < 0.05 and ** *p <* 0.01 versus 5-FU group.

**Figure 3 pharmaceuticals-17-00297-f003:**
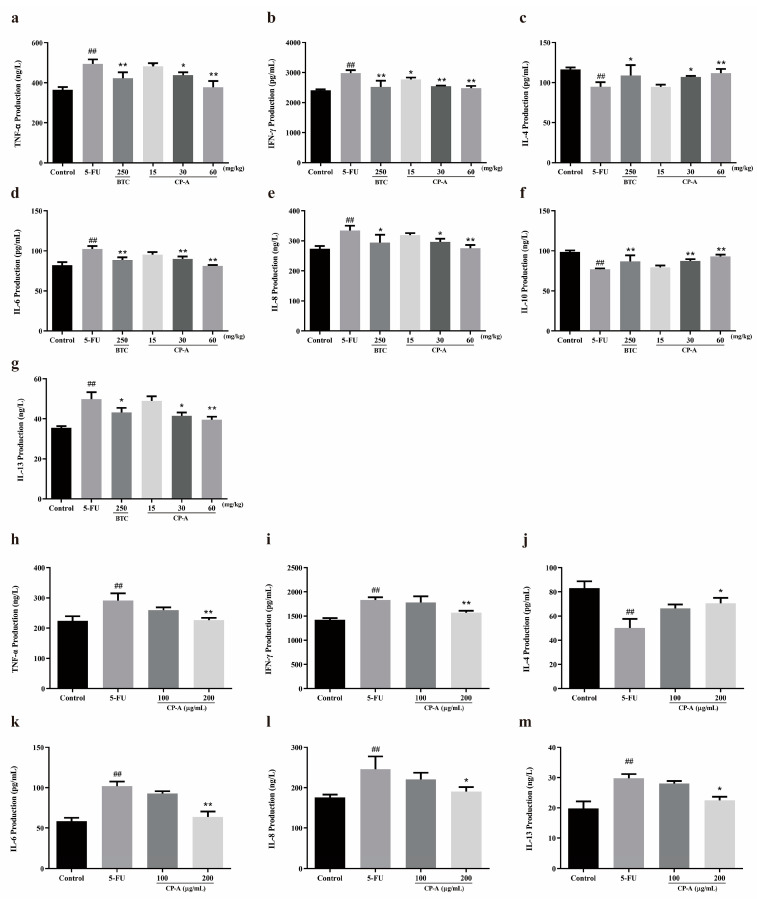
Effects of CP-A on inflammatory cytokines in 5-FU induced rats and IEC-6 cells: (**a**–**g**) The levels of TNF-α, IFN-γ, IL-4, IL-6, IL-8, IL-10 and IL-13 in rat serum (n = 6). (**h**–**m**) The levels of TNF-α, IFN-γ, IL-4, IL-6, IL-8 and IL-13 in cellular supernatant (n = 6). Values are represented as mean ± SEM; ## *p* < 0.01 versus control group, * *p* < 0.05 and ** *p <* 0.01 versus 5-FU group.

**Figure 4 pharmaceuticals-17-00297-f004:**
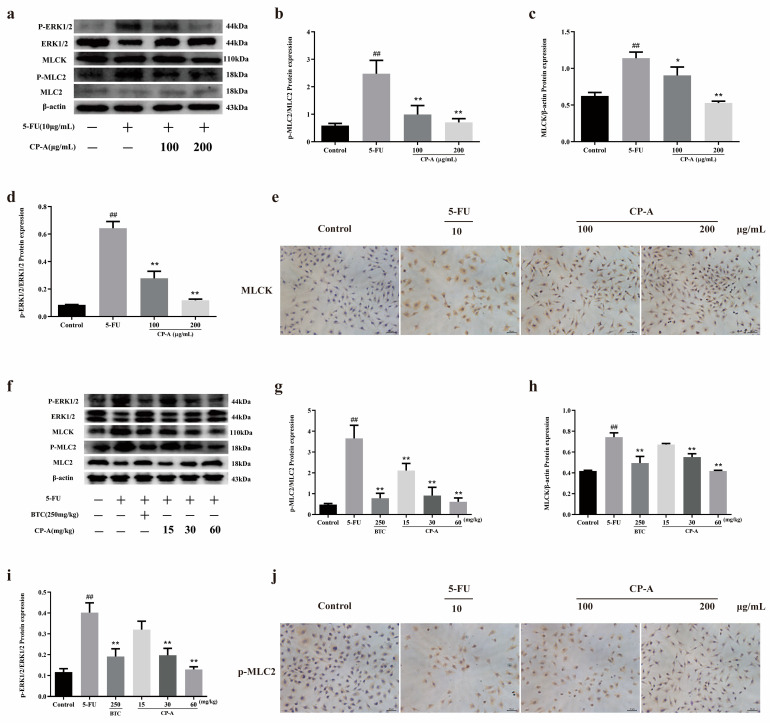
Effects of CP-A on ERK/MLCK/MLC2 signaling pathway proteins in 5-FU-induced rats and IEC-6 cells: (**a**–**d**) Expression of MLC2, p-MLC2, MLCK, ERK1/2 and p-ERK1/2 in IEC-6 cells. (**e**) MLCK protein in IEC-6 cells (magnification 100×). (**f**–**i**) Expression of MLC2, p-MLC2, MLCK, ERK1/2 and p-ERK1/2 in intestinal. (**j**) The p-MLC2 protein in IEC-6 cells (magnification 100×). Values are represented as mean ± SEM (n = 3); ## *p* < 0.01 versus control group, * *p* < 0.05 and ** *p <* 0.01 versus 5-FU group.

**Figure 5 pharmaceuticals-17-00297-f005:**
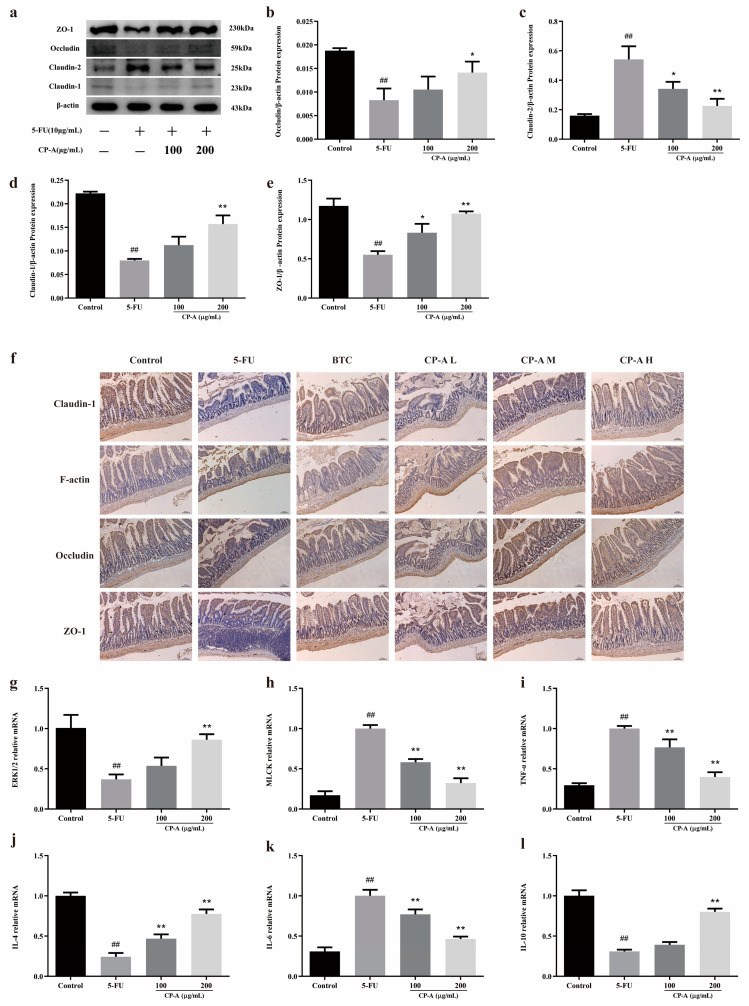
Effects of CP-A on intestinal mucosal barrier proteins: (**a**–**e**) Expression of Claudin-1, Claudin-2, Occludin and ZO-1 proteins in IEC-6 cells. (**f**) Expression of Claudin-1, F-actin, Occludin and ZO-1 proteins in intestinal (magnification 50×). (**g**–**l**) Relative mRNA expression of ERK1/2, MLCK, TNF-α, IL-4, IL-6 and IL-10. Values are represented as mean ± SEM (n = 3); ## *p* < 0.01 versus control group, * *p* < 0.05 and ** *p <* 0.01 versus 5-FU group.

**Figure 6 pharmaceuticals-17-00297-f006:**
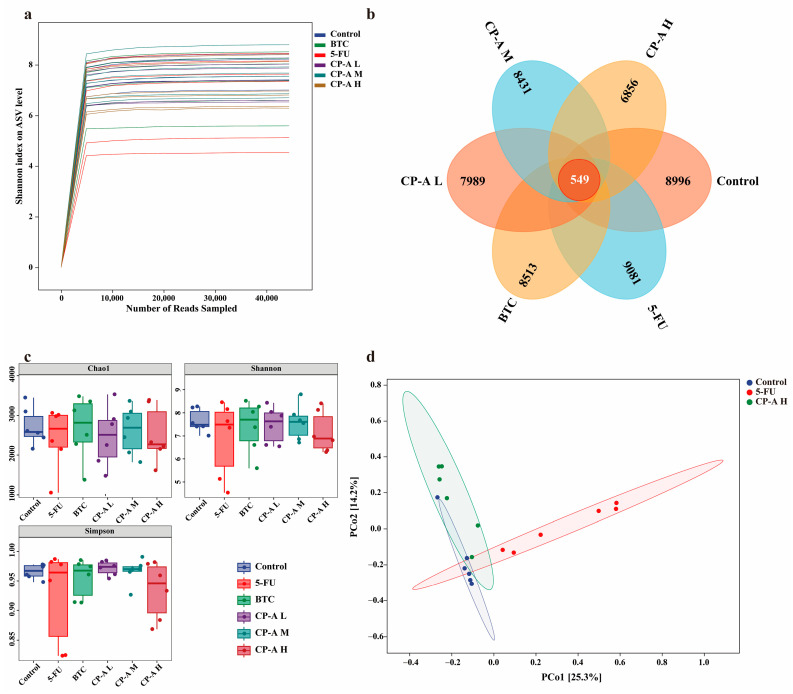
Effects of CP-A on ASV and microbial diversity analysis: (**a**) Rarefaction curve of Shannon index. (**b**) Venn diagram of ASVs. (**c**) Alpha diversity (Chao1 index, Shannon index and Simpson index); (**d**) Beta diversity (Bray–Curtis). Values are represented as mean ± SEM (n = 6).

**Figure 7 pharmaceuticals-17-00297-f007:**
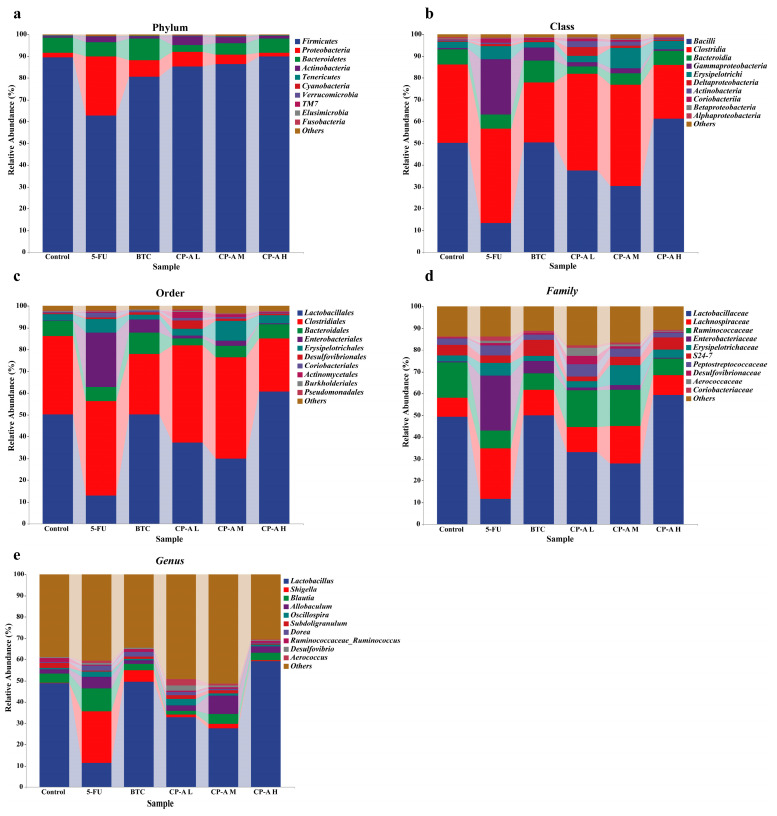
Taxonomy analysis of microbiota communities at the Phylum, Class, Order, Family and Genus levels (n = 6). (**a**) Phylum levels, (**b**) Class levels, (**c**) Order levels, (**d**) Family level, (**e**) genus level.

**Figure 8 pharmaceuticals-17-00297-f008:**
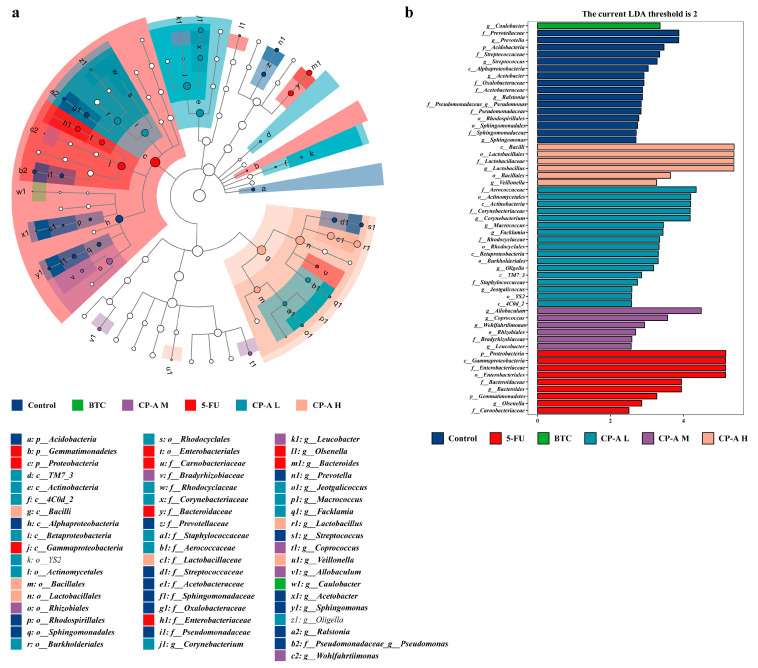
LEfSe analysis (n = 6): (**a**) Each group of LEfSe analysis by cladogram. (**b**) LDA distribution histogram of significantly different species in each group.

## Data Availability

The cecum microbiota gene sequence dataset presented in this study has been made available at the NCBI online repository: NCBI, Submission ID: SUB14100411, BioProject ID: PRJNA1054780. “C” represents the control group, “M” represents the 5-FU group, “Y” represents the BTC group, “D” represents the CP-A L group, “Z” represents the CP-A M group and “G” represents the CP-A H group.

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
