# Peer review of "An Inulin-Type Fructan CP-A from Codonopsis pilosula Alleviated 5-Fluorouracil-Induced Intestinal Mucositis via the ERK/MLCK/MLC2 Pathway and Regulation of Gut Microbiota"

_pharmaceuticals, 2024, doi:10.3390/ph17030297_

Round 1

Reviewer 1 Report

Comments and Suggestions for Authors

This manuscript assesses the use of Codonopsis pilosula-derived CP-A as a tool to mitigate Intestinal mucositis (IM) and its mechanism using in vitro and rat models. They used Western blot, immunohistochemical (IHC) and real-time PCR (RT-PCR) analyses to assess protein expression related to the extracellular-regulated protein kinases (ERK)/myosin light chain kinase (MLCK)/myosin light chain 2 (MLC2) signaling pathway and tight junction proteins.

Comments

Lin 128- . Total RNA Extraction and Quantitative Real-Time PCR

This section needs to be more clear, authors can use this reference as a guide for that (Madkour, Mahmoud, et al. "Early life thermal stress modulates hepatic expression of thermotolerance related genes and physiological responses in two rabbit breeds." Italian Journal of Animal Science 20.1 (2021): 736-748.

·         Why did use GAPDH as a house keeping gene , they should used two reference genes like beta actin also.

·         Line 164: authors mentioned that . Small-intestinal tissues were rapidly harvested and rinsed with cold. Which segment in the small intestine????

·         Why did they use ileum, not jejunum?

Reviewer 2 Report

Comments and Suggestions for Authors

1.     Check the list of authors, seems some authors are omitted.

2.     Methodology: The methodology related to the analysis of gut microbiota is vague. R is a program not a package. And R is usually use do display the statistical analysis of the data. The use of QIIME2 for processing the sequence reads should be elaborated.

3.     Line 233: Correct the heading, “Results”.

4.     Results & Discussion: What could be the possible reason for the decrease of Lactobacillus and increase of Shigella in the organisms treated with 5-FU? Moreover, the genus level variations between the control and test groups should be discussed more in details with possible mechanisms.

5.     The resolution of the Figure 7 is very low.

6.     Heading 3.8 is misleading. Authors have mentioned proteins while discussing variation in bacterial groups.

7.     Authors should elucidate the mechanisms of the CP-A on the rats by focusing more up to protein level (Expression level). 

Reviewer 3 Report

Comments and Suggestions for Authors

This article deals with an exciting approach to analyzing gut microbiota dysbiosis and oncological treatments. The authors have a good research question. However, this manuscript lacks transparency related to the raw sequences of 16s amplicon sequencing and their pipelines and bioinformatic scripts.

Detailed comments and suggestions are in the file I've attached; please revise them.

Comments on the Quality of English Language

Moderate editing of English language required

Round 2

Reviewer 2 Report

Comments and Suggestions for Authors

The authors have addressed the reviewer's concerns. It can be accepted for publication in its current version. 

Author Response

Thanks for your consideration

Reviewer 3 Report

Comments and Suggestions for Authors

The authors need to unlock their bio project at NCBI because their IDs actually are not working (NCBI; Submission ID: SUB14100411; BioProject ID: PRJNA1054780)

Comments on the Quality of English Language

 Minor editing of English language required

Author Response

Thanks for your consideration,the bio project at NCBI has been unlocked.

Round 3

Reviewer 3 Report

Comments and Suggestions for Authors

Authors need to add supporting information to identify their samples at NCBI correctly. It is not clear how samples are from "control," "BTC," "CP-AM," etc.

Comments on the Quality of English Language

Minor editing of English language required

Author Response

1. Authors need to add supporting information to identify their samples at NCBI correctly. It is not clear how samples are from "control," "BTC," "CP-AM," etc.

Response:Thanks for your valuable suggestion. We have supplemented related information in the section of “Availability of data and materials”. “C” represents the control group, “M” represents the 5-FU group, “Y” represents the BTC group, “D” represents the CP-A L group, “Z” represents the CP-A M group, and “G” represents the CP-A H group.

2. Minor editing of English language required

Response:Thanks for your comment. The English grammar, punctuation and spelling of our manuscript have been carefully checked and revised by a native speaker to minimize the grammatical errors and incorrect writing. The revised portion has been highlighted with yellow color for indication.